# MiR-23b Promotes Porcine Preadipocyte Differentiation via *SESN3* and *ACSL4*

**DOI:** 10.3390/cells11152339

**Published:** 2022-07-29

**Authors:** Meng Li, Na Zhang, Jiao Li, Wanfeng Zhang, Wei Hei, Mengting Ji, Yang Yang, Guoqing Cao, Xiaohong Guo, Bugao Li

**Affiliations:** College of Animal Science, Shanxi Agricultural University, Taigu 030801, China; 13994576150@163.com (M.L.); 15135402330@163.com (N.Z.); jiaoli20201026@163.com (J.L.); zhangwanfeng123@126.com (W.Z.); heiwei1234@126.com (W.H.); jimengting2864@163.com (M.J.); yangyangyh@sxau.edu.cn (Y.Y.); anniecao710502@aliyun.com (G.C.)

**Keywords:** pig, miR-23b, *SESN3*, *ACSL4*, preadipocyte, adipogenic differentiation

## Abstract

Genetic improvement of pork quality is one of the hot topics in pig germplasm innovation. Backfat thickness and intramuscular fat content are important indexes of meat quality. MiRNAs are becoming recognized as a crucial regulator of adipose development. Therefore, it is crucial to understand how miR-23b regulates fat metabolism at the molecular level. In the present study, Oil Red O staining, and Western blot were used to evaluate the effect of miR-23b on the differentiation of porcine preadipocytes. Dual-luciferase reporter gene assay, pulldown, and RIP were used to reveal the mechanism of miR-23b regulating cell differentiation. The findings demonstrated that miR-23b promotes the expression of adipogenic factors and increases the content of lipid droplets, thus promoting the differentiation of preadipocytes. Further research found that miR-23b can directly bind to the 3’UTR of *SESN3* to regulate adipogenic differentiation. In addition, it was speculated that miR-23b controls cell differentiation by positively regulating the expression of *ACSL4* in other ways. Here, we demonstrate that miR-23b promotes the differentiation of porcine preadipocytes by targeting *SESN3* and promoting the expression of *ACSL4*. The present study is meaningful to the improvement of pork quality and the development of animal husbandry.

## 1. Introduction

Pork is one of the main sources of animal protein for humans and a crucial component of animal husbandry’s economic underpinning. People have more exacting standards for the quality of pork as their standard of living rises. Therefore, improving the quality of pork has become an urgent problem for pig breeders. Backfat thickness and intramuscular fat content are closely related to pork quality [1,2,3]. Therefore, fully understanding the molecular mechanism behind the adipocyte proliferation and differentiation is of great significance to improve the economic value of animal husbandry. However, the precise molecular processes that underlie lipid metabolism are largely unknown.

Currently, studies on the regulatory factors of adipogenic differentiation mainly focus on some transcription factors, including PPARγ [4,5], C/EBPβ [6,7], FABP4 [8], etc. In recent years, it has been found that miRNAs, which are typically noncoding RNAs of 19–24 nt [9], have a significant role in regulating the fat metabolism [10,11,12]. MiR-370 was found to promote the cell proliferation, while suppressing the differentiation of porcine preadipocytes [13]. Liu et al. demonstrated that miR-375 negatively regulates the preadipocyte adipogenic differentiation by targeting bone morphogenetic protein receptor type 2 in pigs [14]. Zhang et al. proved that miR-181a promotes porcine preadipocyte adipogenic differentiation by sponging *TGFBR1* [15]. As a result, miRNA is crucial for the differentiation of porcine preadipocytes. However, several more regulatory miRNAs that are involved in pig adipogenesis remain largely unknown.

Numerous studies have shown that miR-23b is crucial for the development of illnesses related to fat metabolism. Mohammad et al. certified that miR-23b leads to fat accumulation by suppressing the expression of Sirtuin 1 (SIRT1) in HepG2 Cells [16]. Li et al. indicated that miR-23b plays a significant role in liver disease by affecting Acyl-CoA thioesterases 4, which ameliorates nonalcoholic steatohepatitis [17]. Liu et al. found that hepatocellular carcinoma (HCC) patients with a high body-fat percentage have a higher expression of miR-23b than HCC patients with a low body-fat percentage [18]. Yet, there is no information on miR-23b’s impact on porcine preadipocyte differentiation.

The present research aimed to explore the regulation of miR-23b on the differentiation of porcine precursor adipocytes. Bioinformatics technology prediction results showed that sestrin3 (*SESN3*) and Acyl-CoA synthetase long-chain family member 4 (*ACSL4*) are target genes of miR-23b. In the present study, dual-luciferase reporter gene assay, pulldown, and RIP were used to explore the mechanism of miR-23b regulating the differentiation of porcine precursor adipocytes through *SESN3* and *ACSL4*. It provides a theoretical basis for the study of the molecular mechanism of fat metabolism in pigs. Our findings might serve as a theoretical foundation for research on the molecular mechanisms in pig fat metabolism.

## 2. Materials and Methods

### 2.1. Ethics Approval and Sample Collection

The experimental protocols used in this study were approved by the Animal Ethics Committee of Shanxi Agricultural University (Shanxi, China, SXAU-EAW-P002003). Jinfen White pigs (castrated boars) were used as experimental animals in present study, provided by Datong Pig Farm (Shanxi, China). Three 90-day-old Jinfen White pigs (castrated boars) were killed by electric shock and bloodletting. After that, longissimus dorsi muscle, heart, spleen, kidney, liver, lung, and subcutaneous fat were collected. In addition, nine Jinfen White pigs (castrated boars) at 1, 90, and 180 days of age were selected, 3 pigs per age. After being killed by electric shock and bloodletting, subcutaneous fat was collected. The tissues were instantly frozen in liquid nitrogen and afterward kept in a −80 °C refrigerator.

### 2.2. Cell Culture and Transfection

Back and neck subcutaneous adipose tissue from the piglets at 7 days of age was taken and placed in sterile Petri dishes, then washed, and all ingredients were removed except fat. The tissue was divided up and digested using 2 mg/mL type I collagenase (Gibco, Waltham, MA, USA) for 1 h. Then, the digestion was terminated with serum-containing medium. The supernatant was removed after 15 min of centrifugation at 1200 rpm, and then the cells were re-suspended by blowing. Following filtering, a 60 mm Petri dish was inoculated with the centrifuge tube’s filtrate. Complete culture medium, DMEM supplemented with 10% fetal bovine serum (FBS), and 1% penicillin–streptomycin (Gibco, USA) were used to culture the precursor adipocytes. Replace the complete medium with the induction medium (complete medium + 5 μmol·L^−1^ insulin + 1 μmol·L^−1^ dexamethasone + 0.5 mmol·L^−1^ 3-isobutyl^−1^-methylxanthine + 1 μmol·L^−1^ rosiglitazone) once the cells have reached confluence. Every two days, the medium was changed, and on day 4, the medium was changed to maintenance medium (complete medium + 5 μmol·L^−1^ insulin). After that, change the medium every two days until the fat droplets fuse into large fat droplets. HEK293T cells were cultured in complete medium. When the degree of cells reached 70%, the transfection reagent Lipofectamine 3000 (Thermo Fisher Scientific, Waltham, MA, USA) was used to transfect the cells, and the complete medium was replaced 6 h after transfection.

### 2.3. Quantitative Real-Time Polymerase Chain Reaction (qRT-PCR)

Total RNA was extracted in accordance with the instructions of Trizol Reagent Kit (Takara, Shiga, Japan). The RNA concentration and purity were determined by ND-1000 nucleic acid protein analyzer and stored at −80 °C. PrimeScript RT Reagent Kit (Takara, Japan) and miRNA 1st Strand cDNA Synthesis Kit (by stem-loop) (Vazyme, Nanjing, China) were used to synthesize cDNA of mRNA and miRNA. qRT-PCR of mRNA and miRNA were performed using SYBR^®^PrimeScriptTM RT-PCR Kit (Takara, Japan) and miRNA Universal SYBR qPCR Master Mix Kit (Vazyme, China). Then, *18S rRNA* and *U6* were used as reference genes. Primers used in present research were synthesized by Sangon Biotech (Shanghai) Co., Ltd., (Shanghai, China) and their sequences are shown in Appendix A. Each sample was repeated 3 times, and the quantitation data were calculated using the 2^−ΔΔCt^.

### 2.4. Western Blot

Total protein was extracted from the cell samples after lysis, and were denatured at 100 °C for 15 min. SDS-PAGE was performed to electrophoresis. After membrane transfer, sealing, and incubation with primary antibodies, the PPARγ antibody (ABclonal, Wuhan, China), FABP4 antibody (ABclonal, China), and β-actin antibody (Bioss, Shanghai, China) were used for Western blot assay. After the second antibody was incubated, exposure and photography were performed to visualize the proteins.

### 2.5. Oil Red O Stain

On the eighth day of cell differentiation, when there were too many lipid droplets, the medium was removed, rinsed with PBS, and then fixed using 4% paraformaldehyde. The formaldehyde was then discarded and thoroughly cleaned with PBS. Rinse the mixture with 60% isopropyl alcohol for 2 min, discard the mixture, and add the mixture and filtered Oil Red O working solution (saturated Oil Red O dye:distilled water = 3:2) for 2 h. Then, the working solution was discarded and rinsed with PBS, and photographs were taken under an inverted microscope.

### 2.6. Target Genes Prediction of miR-23b

Using miRDB (http://mirdb.org/; accessed on 1 October 2021), TargetScan (http://www.targetscan.org/; accessed on 1 October 2021), and starBase (https://starbase.sysu.edu.cn/starbase2/browseClipSeq.php; accessed on 1 October 2021) to predict the target genes of miR-23b, Cytoscape software was used to construct the miR-23b-mRNA regulatory network. DAVID (https://david.ncifcrf.gov/; accessed on 1 November 2021) website was used to perform GO and KEGG analysis of miR-23b target genes. RNAhybird software (https://bibiserv.cebitec.uni-bielefeld.de/rnahybrid; accessed on 1 February 2022) was used to predict the binding sites of miR-23b and target genes.

### 2.7. Dual-Luciferase Reporter Gene Assay

PsiCHECK2-SESN3-Wt and psiCHECK2-SESN3-Mut vectors were created and manufactured in accordance with the expected binding sites of miR-23b and SESN3 predicted by RNAhybird software. HEK293T cells were laid in cell-culture plates, and Lipofectamine 3000 was used for transfection after the cells were fused to about 70%, including four groups: psiCHECK2-SESN3-Mut + mimics NC, psiCHECK2-SESN3-Wt + miR-23b mimics, psiCHECK2-SESN3-Mut + miR-23b mimics, psiCHECK2-SESN3-Wt + mimics NC. After transfection for 48 h, the cells were washed with PBS, and then the following tests were conducted in accordance with the instructions of Dual-Luciferase Reporter Assay Kit (Promega, Madison, WI, USA). Each assay was repeated eight times. Finally, Renilla and Firefly Luciferase were counted, and finally the ratio of them was calculated.

### 2.8. RNA-Binding Protein Immunoprecipitation Assay (RIP)

A Magna RIP RNA-binding protein immunoprecipitation kit (Millipore, Burlington, MA, USA) was used to conduct RIP assay. AGO2 (Boster, Pleasanton, CA, USA), and IgG antibodies (ABclonal, Wuhan, China) were used in this assay. At the end of the experiment, the magnetic beads were washed with lysis solution, and the RNA bound to the magnetic beads was extracted using TRIzol reagent. Changes in the expression of miR-23b, *SESN3*, and *ACSL4* were detected using qRT-PCR.

### 2.9. miRNA Pulldown

Two 15cm dishes were cultured with about 2 × 10^7^ cells. Biotin-labeled miRNA probes (experimental group) and negative control probes (negative control group) were transfected, respectively. After 48 h of culture, the cells were washed twice with pre-cooled PBS. miRNA pulldown Kit (BersinBio, Guangzhou, China) was used for the miRNA pulldown test. Magnetic bead sealing, cell lysis, hybridization and incubation, elution and precipitation of RNA were carried out in sequence according to the instruction. After the pulldown test was completed, qRT-PCR was used for follow-up detection.

### 2.10. Statistical Analysis

All data are shown as mean ± SEM. Statistical analyses were performed using SPSS 22.0. Student’s *t*-test was used to assess data differences between two groups, and one-way analysis of variance (ANOVA) was used to compare data differences between three or more groups. Significant difference is indicated by the symbols * at the 0.05 level and ** at the 0.01 level.

## 3. Results

### 3.1. Expression Analysis of miR-23b in Pigs

The expression profile analysis revealed that miR-23b was expressed in a variety of pig tissues. Adipose tissue had the highest expression level, followed by the heart, kidney, and skeletal muscle (Figure 1A). At various developmental stages, pig adipose tissue displayed variable expression of miR-23b, which exhibited an upward trend with increasing age (Figure 1B). In addition, miR-23b was differentially expressed during adipogenic differentiation of porcine precursor adipocytes, with the highest expression on day 4 and day 6 for differentiation (Figure 1C). It is speculated that miR-23b may regulate the adipogenic differentiation in porcine precursor adipocytes.

### 3.2. miR-23b Promotes Adipogenic Differentiation of Porcine Precursor Adipocytes

After miR-23b mimics was transfected in porcine precursor adipocytes, the miR-23b expression was dramatically raised (Figure 2A, *p* < 0.01), which could be used in subsequent experiments. On day 8 of differentiation, the mRNA and protein levels of adipogenic factors were dramatically improved in the miR-23b mimics group compared with the control group (Figure 2B,C, *p* < 0.01). Oil Red O staining showed that miR-23b could promote lipid-droplet formation (Figure 2D). After miR-23b inhibitor transfection, the effects were reversed. These findings suggest that miR-23b promotes the adipogenic differentiation of porcine precursor adipocytes.

### 3.3. Construction of miR-23b-mRNA Network

Combining the prediction results of miRDB, TargetScan, and starBase online software, the miR-23b-mRNA network was constructed (Figure 3A). The predicted target genes include several genes that regulate fat metabolism, such as *ACSL4*, *SIRT1*, *SENS3*, etc. (Figure 3B). GO and KEGG analysis of the target genes revealed that they were mainly enriched in some signaling pathways related to muscle development and fat metabolism, such as the MAPK, FoxO, p53, etc. (Figure 3C,D).

### 3.4. miR-23b Regulates Cells Adipogenic Differentiation by Binding SESN3

*SESN3* has a strong binding force with miR-23b and can control fat metabolism, thus, the present research selected *SESN3* gene for further study. As shown in Figure 4A, after transfection of miR-23b mimics on porcine precursor adipocytes, the miR-23b expression was extremely markedly upregulated, while the *SESN3* expression was decreased (*p* < 0.05). *SESN3* expression considerably increased the following transfection with the miR-23b inhibitor (Figure 4B, *p* < 0.05). RNAhybird software predicted that the seed sequence of miR-23b could bind to the 3’UTR of the *SESN3* gene (Figure 4C). The binding location of miR-23b and *SESN3* was used to create *SESN3* mutant-type and wild-type vectors (Figure 4D). The results of the dual-luciferase reporter gene test showed that the luciferase activity was dramatically decreased in the transfected psiCHECK2-SESN3-Wt + miR-23b mimics group compared to in the psiCHECK2-SESN3-Mut + miR-23b mimics group (Figure 4E, *p* < 0.05). The lucifase activity of the transfected psiCHECK2-SESN3-Wt + miR-23b mimics group was dramatically lower than that of the psiCHECK2-SESN3-Wt + mimics NC group (Figure 4E, *p* < 0.05). These findings suggest that miR-23b may bind to *SESN3*.

According to the results of the RIP assay, both miR-23b and *SESN3* could bind to AGO2 protein, indicating that miR-23b may bind to AGO2 protein and reduce the expression of *SESN3* (Figure 4F).

Further detection by the biotinase-coupled miRNA pulldown test showed that the miR-23b pulldown group had obvious bands, while the control pulldown group had no obvious bands. These results indicate that *SESN3* can be pulled down by the miR-23b biotin probe. Through qRT-PCR detection and calculation of the proportion of pulldown/input, the SESN3 expression in the miR-23b pulldown group accounted for 0.668% of the input group, while that in the control pulldown group accounted for 0.025% of the input group. This indicates that miR-23b can directly bind to *SESN3* (Figure 4G,H).

### 3.5. miR-23b Restores the Inhibitory Effect of SESN3 on Cells Adipogenic Differentiation

The mechanism of miR-23b regulating adipogenic differentiation through *SESN3* was further investigated by a recovery test. According to the results, compared with the control group, the expression levels of lipogenic-related factors and lipid-droplet production were lower in the OE-SESN3 group (Figure 5). These results suggest that *SESN3* does inhibit the adipogenic differentiation of porcine preadipocytes. In addition, miR-23b mimics transfection with overexpressed *SESN3* significantly increasing the expression of adipogenic factors and lipid-droplet production (Figure 5), suggesting that miR-23b can partially relieve the inhibition of *SESN3* on adipogenic differentiation.

### 3.6. Regulatory Relationship between miR-23b and ACSL4

After transfection, miR-23b mimics in porcine precursor adipocytes, and the *ACSL4* expression was increased (Figure 6A, *p* < 0.05), while the *ACSL4* expression was decreased after transfection by the miR-23b inhibitor (Figure 6B, *p* < 0.05). miR-23b expression was positively correlated with *ACSL4*. RNAhybrid software predicted that the seed sequence of miR-23b could bind to the 3’UTR of *ACSL4* with a free energy of −15.8 kcal/mol, showing a weak binding ability (Figure 6C). In addition, the seed sequence of miR-23b can also bind to the CDS region of *ACSL4* with a free energy of −26.8 kcal/mol, indicating a strong binding ability (Figure 6D). Further, the results of pulldown showed that miR-23b could bind to *ACSL4*, but the input was 0.195%, less than 0.5%, indicating a weak binding or non-binding ability (Figure 6E,F). These findings imply that miR-23b may be important in controlling adipogenic differentiation of porcine precursor adipocytes, by positively regulating the expression of *ACSL4* in other ways. Further research is needed about this part.

## 4. Discussion

Further study of the mechanism of fat metabolism is meaningful to the development of animal husbandry. In addition, porcine adipocytes are also employed as a model to investigate human metabolic illnesses since Homo sapiens and Sus scrofa have many similar physiological traits. In present study, we focused on the molecular mechanism of porcine fat metabolism and demonstrated that miR-23b promotes the differentiation of porcine preadipocytes and participates in the regulation of adipogenic differentiation by directly binding to *SESN3*. Therefore, a thorough understanding of the molecular mechanisms behind pig fat metabolism may prove invaluable in improving the financial performance of livestock farms as well as in the battle against diabetes and obesity in humans.

Previous studies have shown that miR-23b is important in the metabolism of lipids and glucose [19]. MiR-23b expression is enhanced in obesity and is connected to adiposity and a higher body mass index [20,21]. Bioinformatics techniques predicted that miR-23b has multiple target genes related to fat metabolism, including *SIRT1*, *ACSL4*, *SESN3*, etc. In addition, previous research has proved that miR-23b can target the 3’UTR region of *SIRT1* [16,19,22]. Picard et al. proved that sirt1 promotes fat mobilization by repressing PPARγ [23]. Liu et al. demonstrated resveratrol inhibits bovine intramuscular adipocytes adipogenesis, by activating the SIRT1-AMPKα-FOXO1 signaling pathway [24]. Moreover, sirt1 may reduce porcine preadipocytes proliferation and differentiation by repressing the expression of FoxO1 [25]. It is speculated that miR-23b may regulate pig fat metabolism and promote the adipogenic differentiation of pig preadipocytes. In the present study, miR-23b was found to promote the adipogenic differentiation of porcine precursor adipocytes, which was in agreement with the results of previous studies. Moreover, this study proved that miR-23b can directly bind to *SESN3* to regulate the differentiation of porcine precursor adipocytes.

Sestrins (SESNs), a family of evolutionarily conserved proteins, play a key role in controlling metabolic balance. SESN3 is a member of the SESNs family, along with members Sestrin1 and Sestrin2 [26,27]. By controlling the AMPK-mTOR axis, SESNs preserve metabolic balance and prevent the metabolic syndrome linked to aging and obesity [28,29]. SESN2 plays a key role in maintaining metabolic homeostasis in animals and preventing various diseases caused by metabolic disorders [29,30,31]. Overnutrition induces high expression of SESN2 in the liver of mice. SESN2 inhibits fat deposition and maintains metabolic homeostasis in liver [29]. In addition, it was found that SESN2 may inhibit adipogenesis by inhibiting JNK expression [32]. *SESN3* is a stress-sensitive gene, which can control lipid metabolism. Overexpression of *SESN3* can improve the accumulation of triglycerides. Triglyceride accumulation was significantly worse after *SESN3* downregulation in cells [33]. By suppressing the SMADs family, SESN3 has been shown to be involved in TGF-Smads signaling and to protect mice from diet-induced non-alcoholic steatohepatitis [31]. Additionally, recent research demonstrated that *SESN3* suppressed the adipogenesis of porcine preadipocytes by inhibiting their proliferation [32]. Our findings support earlier research indicating that *SESN3* prevents preadipocyte development in pigs.

The ACSL4 plays an important role in lipid synthesis and fatty acid degradation, which transforms free long-chain fatty acids into fatty acyl-CoAesters [34,35]. Chen et al. found that the ACSL4 promoted the expression of SREBP1 by c-Myc, thus increasing the triglycerides, cholesterols, and lipid droplets in HCC cells [36]. ACSL4 is closely related to ferroptosis, and its abnormal expression can lead to hepatocellular carcinoma [37], breast cancer [38] and other diseases. Belkaid et al. demonstrated that ACSL4 can inhibit the adipogenic differentiation of macrophages [39]. In addition, ACSL4 also plays an important role in fat metabolism in pigs. Previous studies have found that *ACSL4* is one of the most often identified potential genes related to intramuscular fat content in pigs [40,41,42,43]. ACSL4 is a positive regulator of adipogenic different cells [44,45,46] and can promote adipogenic differentiation of porcine intramuscular precursor adipocytes [44,47]. 

The present study proved that both miR-23b and ACSL4 can promote cell adipogenic differentiation, and their expression levels were positively regulated. MiRNAs regulate gene expression in post-transcriptionally through mRNA degradation or translation suppression [48]. Target genes and miRNA expression levels are negatively regulated, and their functions are generally opposite. In addition, Liang et al. found that miR-339 selectively stimulates GPER1 to inhibit the growth of breast cancer cells [49]. Fat metabolism regulation is a complex process in which various genetic factors interact to maintain metabolic homeostasis. Therefore, we speculated that miR-23b might promote the differentiation of porcine preadipocytes, by positively regulating the expression of *ACSL4* in some way. As for the molecular mechanism of miR-23b that regulates porcine precursor adipocytes, our research group is still conducting in-depth studies.

## 5. Conclusions

The present study demonstrated that miR-23b promotes adipogenic differentiation of porcine precursor adipocytes (Figure 7). Mechanistic studies have shown that miR-23b reduces *SESN3* expression by binding to the 3 ‘UTR of *SESN3* to promote adipogenesis. In addition, miR-23b might promote the differentiation of porcine preadipocytes, by positively regulating the expression of *ACSL4* in some way. Our results lay a theoretical foundation for understanding the role and mechanism of miR-23b in regulating adipogenic differentiation in pigs, which may be of significance for improving pork quality.

## Figures and Tables

**Figure 1 cells-11-02339-f001:**
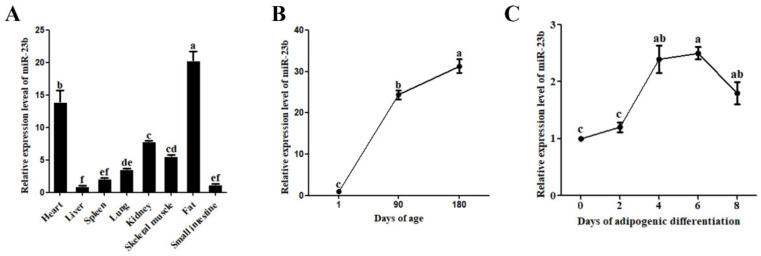
Expression patterns of miR-23b. (**A**) Expression analysis of miR-23b in various tissues of pigs; (**B**) expression analysis of miR-23b in pig adipose tissue at different developmental stages; (**C**) expression analysis of miR-23b during adipogenic differentiation of porcine precursor adipocytes. Note: Different lower-case letters of shoulder label indicated significant difference at 0.05 level.

**Figure 2 cells-11-02339-f002:**
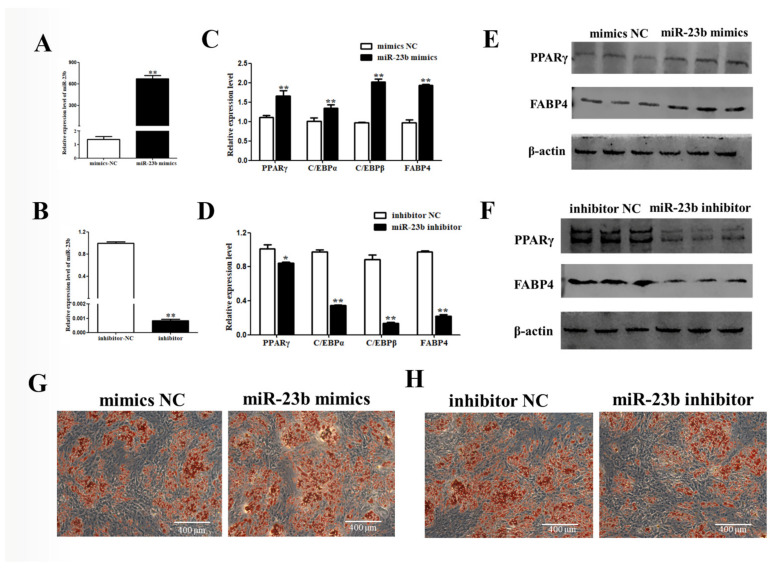
miR-23b promotes adipogenic differentiation of porcine precursor adipocytes. (**A**,**B**) Cell transfection efficiency; (**C**,**D**) expression changes of adipogenic factors at mRNA level; (**E**,**F**) expression changes of adipogenic factors at protein level; (**G**,**H**) the results of Oil Red O staining. Note: * indicted significant difference at 0.05 level, and ** indicted significant difference at 0.01 level.

**Figure 3 cells-11-02339-f003:**
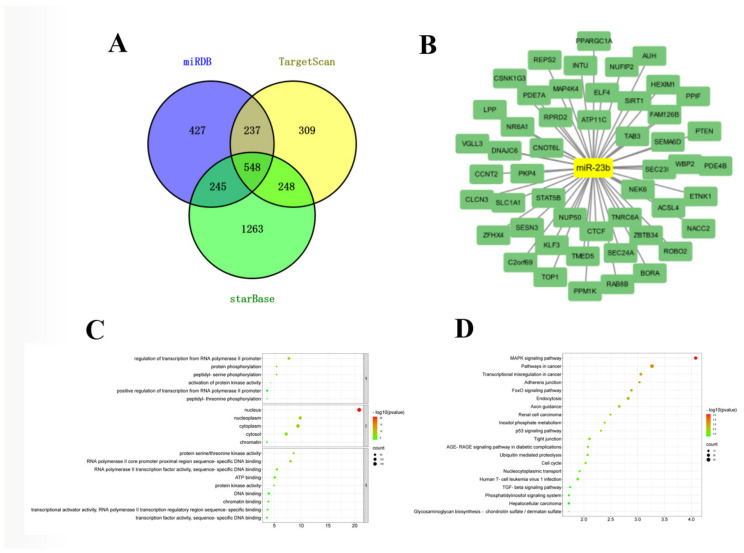
Prediction of target genes of miR-23b and functional enrichment analysis of target genes. (**A**) Online software predicted the number of target genes of miR-23b; (**B**) miR-23b-target genes network; (**C**,**D**) the results of GO and KEGG of target genes.

**Figure 4 cells-11-02339-f004:**
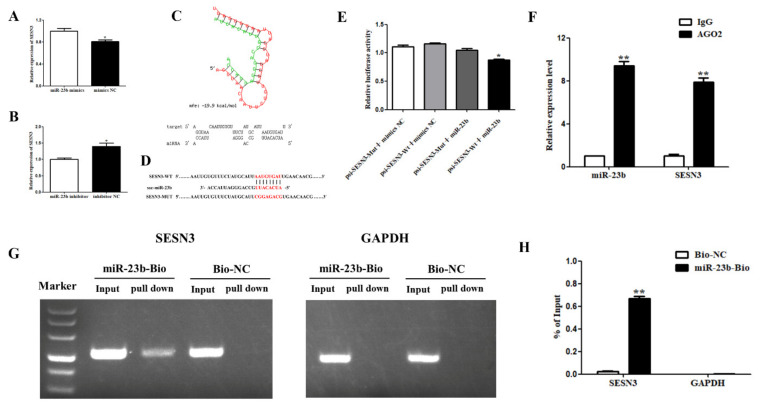
*SESN3* serve as a miR-23b sponge. (**A**,**B**) Expression changes of *SESN3* following transfection with miR-23b mimics or inhibitor; (**C**) the predicted results of binding sites of *SESN3* and miR-23b; (**D**) the sequence of psiCHECK2-circIGF1R-Wt and psiCHECK2-circIGF1R-Mut; (**E**) the results of dual-luciferase reporter assay; (**F**) the results of AGO2-RIP assay; (**G**,**H**) the results of miR-23b pulldown. Note: * indicted significant difference at 0.05 level, and ** indicted significant difference at 0.01 level.

**Figure 5 cells-11-02339-f005:**
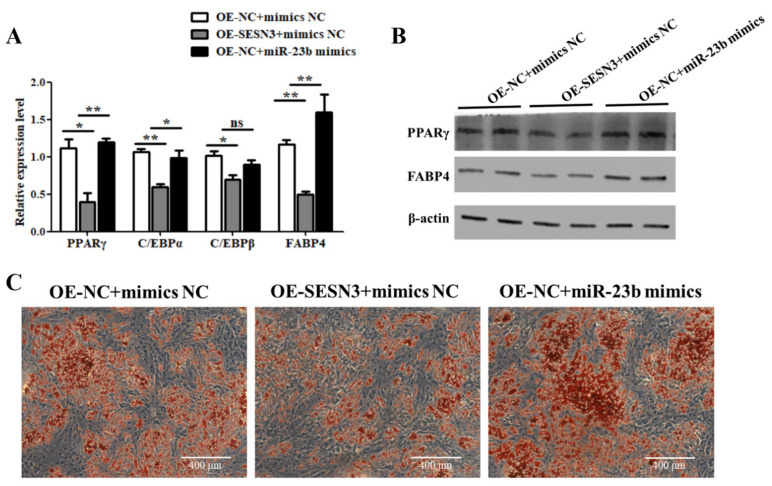
Recovery experiment of miR-23b and *SESN3*. (**A**,**B**) The expression changes of adipogenic marker genes; (**C**) the results of Oil Red O staining. Note: * indicted significant difference at 0.05 level, and ** indicted significant difference at 0.01 level.

**Figure 6 cells-11-02339-f006:**
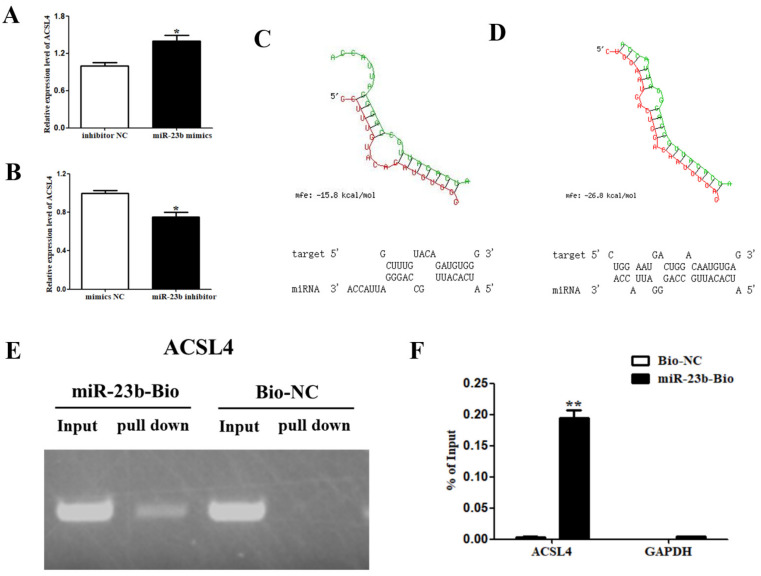
Regulatory relationship between miR-23b and *ACSL4*. (**A**,**B**) *ACSL4* expression changes following transfection with miR-23b mimics or inhibitor; (**C**,**D**) RNhybrid was used to predict binding sites of *ACSL4* and miR-23b; (**E**,**F**) the results of miR-23b pulldown. Note: * indicted significant difference at 0.05 level, and ** indicted significant difference at 0.01 level.

**Figure 7 cells-11-02339-f007:**
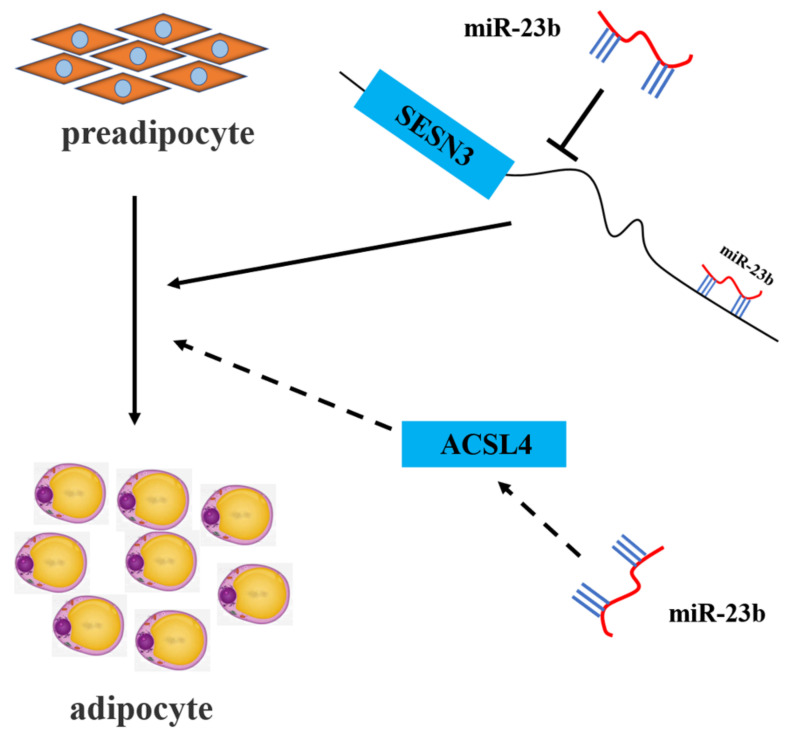
Schematic illustration of the mechanism of miR-23b regulating fat metabolism in pigs.

## Data Availability

Not applicable.

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
