# Peer review of "MiR-23b Promotes Porcine Preadipocyte Differentiation via SESN3 and ACSL4"

_cells, 2022, doi:10.3390/cells11152339_

Round 1

Reviewer 1 Report

The paper provides a new information concerning the role of miR23b in adipocyte differentiation in pigs. In paper Authors put a lot of interest into new protein family senstrins. 

In my opinion Author should improve the Introduction, paper deals with molecular mechanism of adipose tissue differentiation, so, information about pig husbandry is not necessary. Instead more information about senstrins and ACSL4 must be put. Moreover, in methods section, I cannot find information about number of animals  used in study, sex, and when exactly the animals and how were euthanized. 

Please, put more information how you collect tissue etc. 

Reviewer 2 Report

Solid study that shows involvement of mir-23b in the regulation of adipocytic differentiation. Authors performed a set of experiments that demonstrate at least one of potential mechanism how mir-23b could influence development of adipocytes. Results are strong, presented results are valid and definetely improve our knowledge of mechanisms of adipocytic differentiation. The work also has practical significance. 

I recommend accepting the manuscript for publication after minor spelling/grammar check.
